# Optimized Scheme for Accelerating the Slagging Reaction and Slag–Metal–Gas Emulsification in a Basic Oxygen Furnace

**Zichao Yin [1,2]**, **Jianfei Lu [3,]***, **Lin Li [1,2]**, **Tong Wang [1,2]**, **Ronghui Wang [1,2]**, **Xinghua Fan [1,2]**, **Houkai Lin [1,2]**, **Yuanshun Huang [4]** and **Dapeng Tan [1,2,]***

[1]  College of Mechanical Engineering, Zhejiang University of Technology, Hangzhou 310014, China; yinzichao@zjut.edu.cn (Z.Y.); linli@zjut.edu.cn (L.L.); wangtong@zjut.edu.cn (T.W.); ronghuiwang@zjut.edu.cn (R.W.); fanxinghua@zjut.edu.cn (X.F.); linhoukai@zjut.edu.cn (H.L.)
[2]  Key Laboratory of E & M, Ministry of Education & Zhejiang Province, Hangzhou 310014, China
[3]  Zhejiang Chendiao Machinery Co. Ltd., Lishui 321404, China
[4]  Automation Engineering Co., Maanshan Iron & Steel Co. Ltd., Maanshan 243000, China; huangyuanshun@126.com
[*]  Correspondence: zjsd@zjsd.com (J.L.); tandapeng@zjut.edu.cn (D.T.)

**Abstract:** Basic oxygen furnace (BOF) steelmaking is widely used in the metallurgy field. The slagging reaction is a necessary process that oxidizes C, Mn, Si, P, S, and other impurities and therefore directly affects the quality of the resultant steel. Relevant research has suggested that intensifying the stirring effect can accelerate the slagging reaction and that the dynamic characteristics of the top blow are the key factor in exploring the related complex physical and chemical phenomena. To address the issue, the standard k-$\omega$ turbulence model and level-set method were adopted in the present work and a fluid dynamics model was developed for a BOF. Accordingly, the slag–metal–gas emulsion interaction and stirring effect were investigated, and the interference mechanism of a multi-nozzle supersonic coherent jet was revealed. Finally, a self-adjustment method based on fuzzy control is proposed for the oxygen lance. The results indicate that the transfer efficiency of jet kinetic energy at the gas–liquid interface is the critical factor for the slagging reaction and that multi-nozzle oxygen lances with a certain twisted angle have important advantages with respect to stirring effects and splashing inhibition. The fuzzy control method predicts that the optimal nozzle twist angle is within the range of 7.2° to 7.8°. The results presented herein can provide theoretical support and beneficial reference information for BOF steelmaking.

**Keywords:** basic oxygen furnace; slagging reaction; slag-metal-gas emulsions; multi-nozzle oxygen lance; supersonic coherent jet; fuzzy control method

## 1. Introduction

Basic oxygen furnace (BOF) steelmaking is a complex physical and chemical process used in the metallurgy and smelting fields (Figure 1). It involves various phenomena such as multiphase flow, heat and mass transfer, dissolution, and chemical reactions. In the BOF process, high-pressure oxygen is blown into the molten bath at supersonic speed via an oxygen lance, where it promotes the oxidation of C, Mn, Si, P, and other impurities in the bath; the removal of these impurities determines the steel quality. In industrial production, the oxygen top-blowing process is generally limited to approximately 15 min. However, less than 2% of the energy is transferred from the oxygen lance to the molten bath [1], with most losses occurring at the gas–liquid interface [2]. Therefore, accelerating

the slagging reaction and improving energy efficiency in BOF steelmaking are the key objectives for improving steel production capacity [3].

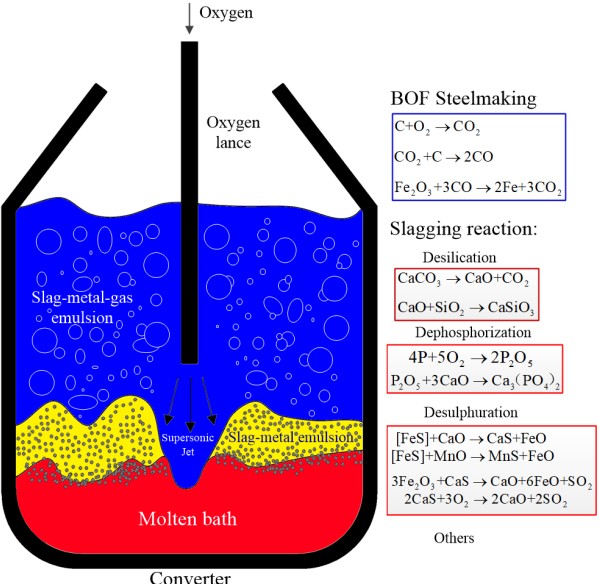

**Figure 1.** Basic oxygen furnace steelmaking and slagging reaction.

Researchers worldwide have been studying the BOF process to address the aforementioned issues. Cao et al. [4] researched the oxygen–slag interaction using numerical simulations and cold model experiments and discussed the relationship between the blowing number and cavity depth. Z. Li et al. [5] predicted the oxygen top-blowing process conducting standard *k-ε*, realizable *k-ε*, standard *k-ω*, and SST *k-ω* models and suggested that, among these models, the standard *k-ω* model is most accurate for describing BOF processes. M. Li et al. [6] evaluated the transfer characteristics from the oxygen lance to the molten bath and discussed the evolution of momentum, energy, and turbulence; their results indicated that the efficiency of momentum transfer can be improved by reducing the lance height or increasing the operating pressure but at the expense of the efficiency of kinetic energy transfer. Sun et al. [7] confirmed the optimum angle of the oxygen lance to be 16° by comparing the slagging time under different nozzle inclination angles. They used a modified lance in PANSTEEL and achieved significant improvements, in which the average time of early slag formed decreased from 5.25 to 4.1 min, the rate of dephosphorization increased from 80.02% to 82.49%, the life of oxygen lance was raised from 428 heats to 800 heats. The aforementioned literature indicates that the jet process is an important dynamic behavior that triggers complex physical and chemical phenomena. Elucidating the evolution mechanism of the multiphase interface and optimizing the oxygen top-blown process would be highly beneficial in accelerating the slagging reaction.

In this context, a modified multi-nozzle oxygen lance with a twist angle (also known as a nozzle-twisted lance) has been proposed and has attracted extensive attention in the industry because it can induce a considerable tangential velocity component [8] and intensify stirring effects [9,10]. Higuchi et al. studied the blow behavior of nozzle twisted lance and verified that a nozzle twist angle (NTA) of 11.4° was most effective for reducing spillage in BOF steelmaking [11]. Li et al. investigated the effects of the NTA on the swirling flow intensity, slagging characteristics, and blow dynamic parameter distributions [12]. Liu et al. [13] analyzed the stirring effects and flow field characteristics of different lances with NTAs of 4°, 8°, and 12° and confirmed that the 8° oxygen lance resulted in a better dephosphorization effect in a 120 t steel converter. The aforementioned literature implies that the nozzle-twisted lance has substantial advantages for the slagging reaction but that the optimum NTA is affected by the flow field [14]. Current research is mainly based on numerical simulations and water-model experiments to provide theoretical support; a method for determining the optimum NTA

has not been developed. In most of the related studies, the converter geometry, lance height, and nozzle inclination angles (NIAs) are assumed to remain unchanged and numerical and experimental analyses are carried out for different NTAs. However, the lance height is often adjusted in BOF steelmaking, which makes the optimum NTA difficult to determine. Addressing this issue requires exploration of the influence of the slagging reaction and the stirring effect from the oxygen lance height, NIA, and NTA.

To accelerate the slagging reaction and improve energy efficiency under a developmental lance height, fuzzy control methods have been used to realize an oxygen lance self-adjustment strategy. In fuzzy control methods, an effective algorithm is used to deal with highly nonlinear models [15]; these methods have been widely applied in various control systems, including those in the metallurgy, machine, artificial intelligence, automation, and other fields [16]. Fuzzy control methods are similar to the human brain control mechanism and are not restricted by a controlled model [17]. The BOF process involves a series of strong nonlinear problems such as chemical reactions, supersonic flow, and multiphysics coupling, among [18]. The fuzzy control method may be adaptable to the adjustment of a multi-nozzle oxygen lance, and it is also the most feasible scheme for use with small-scale data sets [19]. In summary, a self-adjustment control strategy of the oxygen lance may be applicable for accelerating the slagging reaction and saving smelting energy.

A solution to the stirring effect during the slagging reaction and optimization of design scheme for the multi-nozzle oxygen lance are key scientific objectives of this paper. To advance these goals, this research was conducted as follows. First, on the basis of computational fluid dynamics (CFD) and a fuzzy control method, a reactor model of a molten bath in MASTEEL is configured. Accordingly, the blow behavior and flow field characteristics during the slagging process are analyzed [20]. Then, the evolution of the gas–liquid interface is evaluated, and the stirring effect under different nozzles is discussed. Finally, a method for adjusting multi-nozzle oxygen lances is proposed to achieve better slagging effects [21]. This research can provide technical support for BOF steelmaking and also provide beneficial reference information for engineering applications in the metallurgy [22], chemical industry [23], machine [24], astronautics[25] and other fields [26].

This paper is organized as follows. In Section 2, the fluid model of the BOF process is set up and the boundary conditions are described. In Section 3, the jet dynamics behavior and stirring effect during the slagging process are discussed. In Section 4, an optimized design scheme for a multi-nozzle oxygen lance is proposed to save smelting energy. The conclusions are presented in Section 5.

## 2. Oxygen Blow and Stirring Dynamics Model

### 2.1. Numerical Model of BOF Process

In this research, the $\phi275 \times 5$ multi-nozzle oxygen lance designed in MASTEEL was studied. Optimized multi-nozzle oxygen lance model and top-blown converter steelmaking process are shown in Figure 2. To compare the performance of different schemes, the NTA of different oxygen lances are 0° (conventional nozzle, Figure 2a), 5°, 8°, and 13° (Figure 2b). Then, the numerical calculation was modeled according to the same proportion, and lance's geometrical parameters are listed in Table 1. To improve the convergence of numerical calculation, some details of the oxygen lance and steel converter were not considered.

In the BOF process, multi-nozzle oxygen lance was positioned above the converter for top-blowing. In the computational domain, the upper zone of steel converter is air and the bottom is molten steel (Figure 2c). Except for the pressure outlet at the top, all steel converter surfaces are regarded as walls. Five-nozzles are considered as pressure inlets, which can provide initial kinetic energy for top blowing. The lance height is defined as the distance from the nozzle exit to the steel liquid level in the BOF.

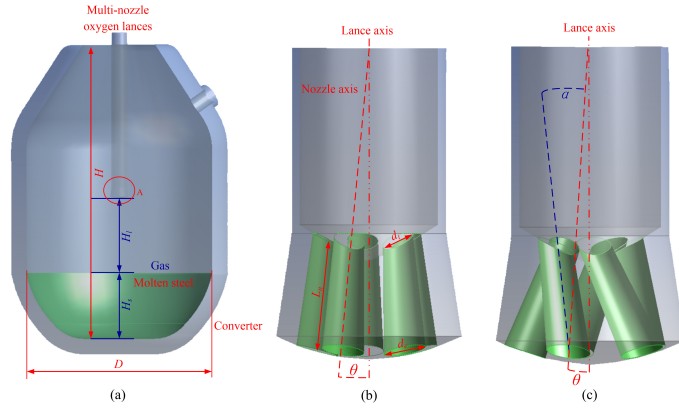

**Figure 2.** Steel converter and multi-nozzle oxygen lances geometrical models: (**a**) oxygen top-blown converter steelmaking; (**b**) magnified view of area A with a conventional oxygen lance and (**c**) a nozzle-twisted lance.

**Table 1.** Geometrical parameters of multi-nozzle oxygen lance and steel converter.

| Geometrical Parameters | Value | Unit |
|---|---|---|
| Nozzle throat diameter ($d_t$) | 35 | mm |
| Nozzle exit diameter ($d_e$) | 48 | mm |
| Nozzle divergent length ($L_n$) | 100 | mm |
| Nozzle inclination angle ($\theta$) | 13.5 | degree |
| Nozzle twist angle ($\alpha$) | 0, 5, 8, 13 | degree |
| Oxygen lance height ($H_l$) | 1100–2100 | mm |
| Design Mach number ($Ma$) | 1.8 | 1 |
| Steel converter diameter ($D$) | 5000 | mm |
| Steel converter height ($H$) | 8000 | mm |
| Steel liquid level ($H_s$) | 2000 | mm |

## 2.2. Governing Equation and Turbulence Model

In the steelmaking process, oxygen and molten steel are regarded as a Newtonian fluid, and the numerical solution can be obtained using the Reynolds averaged Naiver–Stokes (RANs) equation. Therefore, the continuity, momentum, and energy equation can be expressed in the following generalized form:

$$\frac{\partial \rho \Phi}{\partial t} + div\left(\rho u \Phi\right) = div\left(\Gamma_\Phi grad\Phi\right) + S_\Phi \tag{1}$$

where $\Phi$ is a generalized dependent variable; $S_\Phi$ is generalized source term; and $u$ is the velocity vector; $\Gamma_\Phi$ is the generalized diffusion coefficient corresponding to $\Phi$; when $\Phi$ takes different variables, the specific forms of each parameter are shown in Table 2.

**Table 2.** Generalized parameter of governing equation.

| Equation | $\Gamma_\Phi$ | $\Phi$ | $S_\Phi$ |
|---|---|---|---|
| Continuity | 0 | 1 | 0 |
| Momentum | $\mu$ | $u$ | $-\partial p/\partial u + S_{Mu}$ |
| Energy | $\chi$ | $i$ | $-p \cdot divu + \Phi + S_i$ |

The related research indicates that the supersonic gas blow is better described by the standard $k$-$\omega$ model than by other models [5]. It is a two-equation eddy-viscosity turbulence model, and the eddy-viscosity-type expression of the Reynolds stress is

$$\tau_{tij} = 2\mu_t(S_{ij} - S_{nm}\delta_{ij}/3) - 2\rho k\delta_{ij}/3 \tag{2}$$

where $\mu_t$ is the eddy viscosity; $S_{ij}$ is the mean-velocity strain-rate tensor; $\delta_{ij}$ is the Kronecker delta. Eddy viscosity can be regarded as

$$\mu_t = \rho_k / \omega \tag{3}$$

The turbulence kinetic energy $k$ and its specific dissipation rate $\omega$ can be expressed as

$$\frac{\partial(\rho k)}{\partial t} + \frac{\partial}{\partial x_j}\left(\rho u_j k - (\mu + \sigma^* \mu_t)\frac{\partial k}{\partial x_j}\right) = \tau_{tij}S_{ij} - \beta^* \rho \omega k \alpha \tag{4}$$

$$\frac{\partial(\rho \omega)}{\partial t} + \frac{\partial}{\partial x_j}\left(\rho u_j \omega - (\mu + \sigma \mu_t)\frac{\partial \omega}{\partial x_j}\right) = \alpha \frac{\omega}{k}\tau t_{ij}S_{ij} - \beta \rho \omega^2 \tag{5}$$

The constants in this turbulence model are defined as $\alpha = \frac{5}{9}$, $\beta = \frac{3}{40}$, $\beta^* = \frac{9}{100}$, $\sigma = 0.5$, $\sigma^* = 0.5$, and $Prt = 0.9$ [27].

### 2.3. Multiphase Flow Coupling Model

The study of oxygen steelmaking mainly involves investigating the supersonic jet dynamics behavior and multiphase flow coupling. The level set method (LSM) is widely used in multiphase flow research because it can solve the dynamic tracking problem of the mobile phase interface excellently [15]. Luo et al. [28] improved the mass-conserving LSM for detailed numerical simulations of liquid atomization. Balcazar et al. [29] presented a numerical study of the buoyancy-driven motion of single and multiple bubbles using the conservative LSM. Kinzel et al. [30] proposed a level set-based approach that applies uniformly to compressible and incompressible multiphase flows. The LSM employed in the aforementioned investigations can provide important reference information for the study of molten steel splashing and real-time tracking at the two-phase interface.

As the smoothing function of LSM, $\phi(x, t)$ describes the interface between different phases. It shows the distances between various points and the interfaces, and the zero-level set function on the interface boundary is defined as

$$\Gamma = \{x | \phi(x, t) = 0\} \tag{6}$$

The conservation equation about $\phi$ is

$$\frac{\partial \phi}{\partial t} + \boldsymbol{u} \cdot \nabla \phi = 0 \tag{7}$$

where $\boldsymbol{u}$ is the velocity vector of the flow field. The whole flow field is then divided into two domains:

$$\phi(x, t) \begin{cases} < 0 & x \in l_1 \\ = 0 & x \in \Gamma \\ > 0 & x \in l_2 \end{cases} \tag{8}$$

where $l_1$ is the domain of the dispersed phase, $l_2$ is the domain of the continuous phase, and $\Gamma$ is the domain of the boundary. The interface normal vector points to the continuous phase from the dispersed phase, and the interface curvature is

$$n = \frac{\nabla \phi}{|\nabla \phi|} \tag{9}$$

$$\kappa = \nabla \cdot \frac{\nabla \phi}{|\nabla \phi|} \tag{10}$$

On the basis of the continuous dielectric surface $\phi$, the value of fluid density $\rho$ and viscosity $\mu$ in the boundary area can be expressed as

$$\rho = \rho_1 + (\rho_2 - \rho_1)H(\phi) \tag{11}$$

where $H(\phi)$ is a smooth Heaviside function

$$H(\phi) = \begin{cases} 0 & \phi < 0 \\ \dfrac{1}{2} & \phi = 0 \\ 1 & \phi > 0 \end{cases} \tag{12}$$

In the LSM iteration process, the level set function will cease to be an exact distance function even after a single time step. One way for addressing the difficulties is to reinitialize $\phi$ as an exact distance function from the evolving front $\Gamma$ at each time step, which is achieved by solving the following equations:

$$\frac{\partial d}{\partial \tau} = sgn(\phi)(1 - |\nabla d|) \tag{13}$$

$$d(x, 0) = \varphi(x) \tag{14}$$

where $\tau$ is virtual time and $d$ is the distance function and converges to 1. $sgn(\phi)$ is a sgn function,

$$sgn(\phi) = \begin{cases} -1 & \phi < 0 \\ 0 & \phi = 0 \\ 1 & \phi > 0 \end{cases} \tag{15}$$

when $sgn(0) = 0$, $d(x, \tau)$ is equivalent to the zero-level set represented by $\phi(x, t)$, the sgn function $sgn(\phi)$ should be smoothed as

$$sgn_\varepsilon(\varphi_0) = \frac{\varphi_0}{\sqrt{\varphi_0{}^2 + \varepsilon^2}} \tag{16}$$

where $\varepsilon$ is the modified value. The solution process needs to be discretized: First, the space is discretized and projected so that the velocity field does not diverge. Then, the time is discretized, and the reconstructed $\phi(x)$ is initialized, thus improving the accuracy of the solution.

## 2.4. Solution Methods and Boundary Conditions

To obtain the numerical prediction, we established the numerical model of the oxygen top-blown converter steelmaking with the boundary conditions listed in Table 3.

**Table 3.** Boundary conditions of the numerical model items.

| Item | Attribute |
|---|---|
| Inlet | Pressure inlet (0.9 MPa) |
| Outlet | Pressure outlet (0.1 MPa) |
| Wall | No slip wall |
| Reference pressure | $1.01 \times 10^5$ Pa |
| Zone | Oxygen, molten steel |

The calculations were based on the finite volume method (FVM). The coupling processing for pressure and velocity was conducted by the pressure implicit with a splitting of operators algorithm to ensure convergence efficiency [16]. By the pressure staggering option (PRESTO) method, discrete pressure interpolation was performed to avoid a high fluctuation of internal pressure. Momentum and energy adopt the second-order upwind to obtain a precise solution. For the splashing of molten steel, a transient calculation was adopted with a time step of $10^{-4}$ s, which satisfies the

Courant–Friedrichs–Lewy (CFL) condition. The inside of the converter was a two-phase fluid of oxygen and molten steel, and the compressibility of the gas was considered. Blown gas temperature was set as 300 K, and molten steel was 1900 K. Relevant physical properties of two phases are listed in Table 4.

**Table 4.** Material attributions of oxygen and molten steel.

| Physical Property | Oxygen | Molten Steel | Unit |
|---|---|---|---|
| Density | compressible | 7200 | kg/m$^3$ |
| Temperature | 300 | 1900 | K |
| Dynamic viscosity | $1.9 \times 10^{-5}$ | $6.5 \times 10^{-3}$ | Pa·S |
| Specific heat capacity | 919 | 670 | J/(kg·K) |
| Heat conductivity | 0.025 | 15.000 | W/(m·K) |

*2.5. Grid-Independence Test*

Mesh quantity greatly affects the accuracy of flow field numerical results, wherein lots of meshes can obtain smaller discrete errors, but easily produce greater round-off errors. To verify the validity and stability of the numerical predictions, the fluid domain was divided using hexahedral grids. Four computational models with different mesh numbers (i.e., case I, case II, case III, and case IV) were tested. The mesh number of four different cases are 245,893 grids, 502,854 grids, 860,254 grids, and 1,035,908 grids. Figure 3 shows the axial velocity distribution at the center of the steel converter in the different cases. It can be found that numerical curves of four different mesh numbers had similar trends. As the mesh quantity reached a certain value, the numerical curves of the velocity took on uniform trends (case III and case IV). However, the variable values (case I) with less mesh quantity are quite a bit lower than that with the other mesh numbers, especially for the prediction on the gas–liquid interface. It was demonstrated that a lower mesh number cannot provide sufficiently accurate results. For the numerical result at the gas–liquid coupling interface, the mesh number had a major impact on the variable predictions. Only slight differences were observed in their peaks and troughs for the curves (case III and case IV), and the numerical error is less than 3.9% (when H = 3.49 m, the velocity of case III is 87.68 m/s, while case IV is 91.23 m/s), which satisfied the accuracy requirement. Therefore, to save computation time, the mesh size consistent with case III was adopted in the present study.

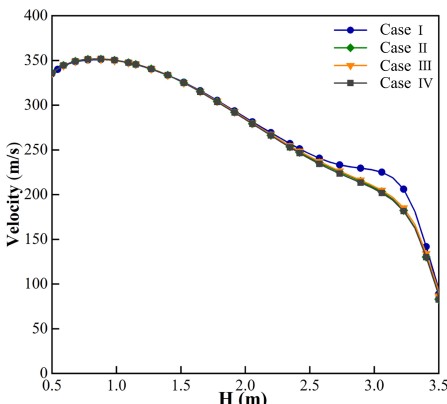

**Figure 3.** Velocity distribution for different mesh sizes.

Since the inlet boundary condition is the pressure inlet, the blown gas in the nozzle will accelerate purely in a short process, but the final blown velocity is approximate to the design Mach number. In the simulation of a conventional oxygen lance, the maximum jet velocity is 475 m/s, about Mach 1.40, and the design Mach number is 1.5. Because the boundary layer is not considered in the design work, the actual jet velocity is slightly lower than the theoretical value, which is similar to our simulation. Therefore, the efflux velocity error of the simulation is within about 5%. It is worth noting that axial

velocity at the center of the steel converter is well below the maximum jet velocity. It is caused by no central nozzle in the designed oxygen lance, and the working five nozzles on the circumference form a low-speed zone at the center.

## 3. Numerical Calculation and Discussion

### 3.1. Analysis of Supersonic Coherent Jets with Multi-Nozzle Oxygen Lance

The analysis of blow characteristics plays a key role in research on oxygen top-blown converter steelmaking. A pressure gradient is the main factor causing the flow field variation, where the dynamic pressure can reflect the kinetic energy characteristics of the jetted gas. On the basis of the LSM, the dynamical pressure profiles of a top-blow in the steelmaking process were obtained (Figure 4). In the first 0.002 s, the oxygen was blown from the nozzles and the molten bath showed no obvious disturbance by the high-speed gas. In Figure 4b, the oxygen with high speed was injected vertically to the molten steel, wherein the dynamic pressure took on a high aggregate state. This produced an intensive shock force that was acting on the gas–liquid interface, as shown in Figure 4c,d. Apparently, the dynamic pressure distribution was stable, and the kinetic energy of oxygen was transferred to the molten bath through the gas–liquid interface, resulting in a series of complex physical phenomena such as dimpling, penetration, and splashing. Thereafter, the dynamic pressure and swing velocity of the molten steel gradually increased and the nozzle-twisted lance began working.

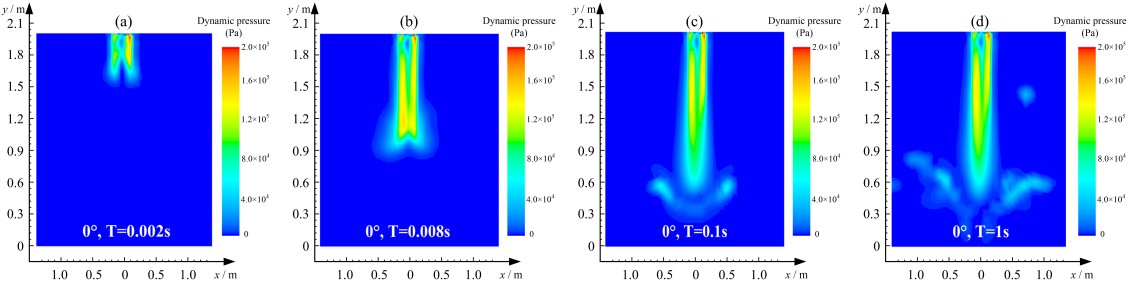

**Figure 4.** Dynamic pressure profiles in the blowing process with multi-nozzle oxygen lance (NTA = 0°). (**a**) Flow time = 0.002 s, (**b**) flow time = 0.008 s, (**c**) flow time = 0.1 s, (**d**) flow time = 1 s.

In Figures 5a and 6a, it can be found that the shock morphology of the oxygen took on a concentration state, which induced the molten steel to be spattered, and the gas–liquid interface had apparent sunken, as shown in Figure 7a,b. However, with the increment of the swirl angle, the dynamic pressure of the oxygen had an apparent decrease, as shown in Figure 5c,d, and the morphology of the shock velocity had changed from concentrated state to the dispersive state. It is indicated that the vertical energy reduces, and circumferential energy increases, as shown in Figure 6c,d. Meanwhile, the spattering phenomenon was vanished, as shown in Figure 7d. To investigate the working efficiency of the nozzle-twisted lance, the physical parameters of the flow field, such as the dynamic pressure, blow velocity, and volume fraction, were compared. Figures 5–7 show the influence of different NTAs on the flow characteristics, especially the dynamic pressure, blow velocity, and two-phase interface evolution. The interference in the dynamic pressure field and velocity field of a conventional nozzle is weak. Therefore, large kinetic energy is generated, which results in a considerable impact depth and area in the molten pool. From the blow characteristics, the swirl effect of the oxygen lance with a 5° NTA is more obvious. Both the dynamic pressure and velocity show substantial swirling motion. Meanwhile, the kinetic energy transfer area of the gas–liquid two-phase interface was relatively concentrated and the top-blowing effect was distinct. By contrast, the impact of the 8° oxygen lance was not substantial. However, the tangential velocity component increased and less kinetic energy was lost at the two-phase interfaces. For the 13° oxygen lance, despite the increase of the NTA, the mutual interference of supersonic coherent jets between the nozzle exits was relatively large, which resulted in a large loss of kinetic energy in the collision between the jets, which in turn

affected the top-blowing efficiency, which is undesirable in oxygen top-blown converter steelmaking. On the basis of the aforementioned discussion, the following four points are inferred:

1.  The dynamic pressure and velocity of the conventional oxygen lance are the largest, and the impact depth is considerable. However, most of the kinetic energy is lost at the two-phase interface.
2.  The 5° nozzle-twisted lance exhibits good performance in both impact kinetic energy and swirl of the blow gas.
3.  The impact kinetic energy of the 8° oxygen lance is weak, whereas the tangential force of molten steel is remarkable, which is more likely to produce swirling.
4.  The greatest kinetic energy dissipation occurs in the 13° oxygen lance blowing processing, and the top-blowing effect is the worst. However, the tangential velocity component also increases correspondingly.

Therefore, the stirring effects of multi-nozzle oxygen lances with different NTAs cannot be judged simply on the basis of the blown characteristics.

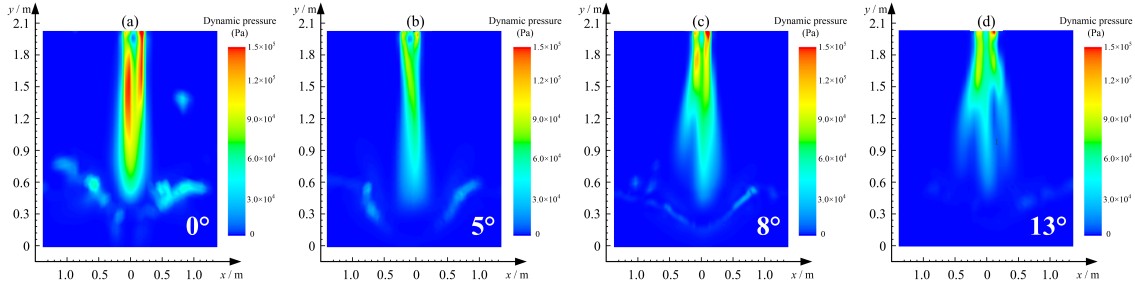

**Figure 5.** Dynamic pressure profiles in top-blowing steelmaking with different multi-nozzle oxygen lances. (**a**) NTA = 0° (**b**) NTA = 5° (**c**) NTA = 8° (**d**) NTA = 13°.

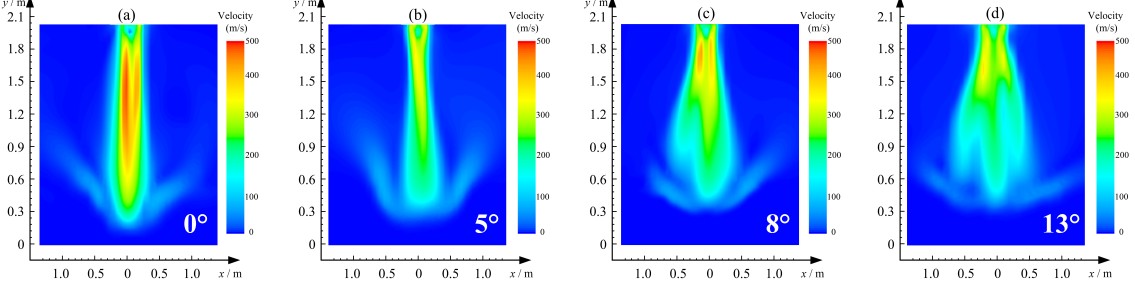

**Figure 6.** Jet velocity profiles in top-blowing steelmaking with different multi-nozzle oxygen lances. (**a**) NTA = 0° (**b**) NTA = 5° (**c**) NTA = 8° (**d**) NTA = 13°.

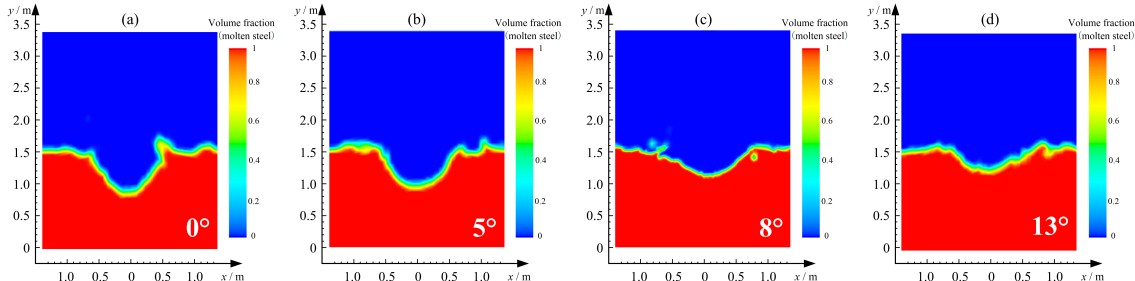

**Figure 7.** Volume fraction profiles in top-blowing steelmaking with different multi-nozzle oxygen lances. (**a**) NTA = 0° (**b**) NTA = 5° (**c**) NTA = 8° (**d**) NTA = 13°.

Turbulence kinetic energy (TKE) is an important reference to measure turbulence intensity and energy transport in a flow field. The distribution of TKE at four different NTAs is shown in Figure 8. From the figure, it can be found that the kinetic energy had the maximum value in the 0° swirl angle.

With the increment of the swirl angle, the vertical kinetic energy has an apparent decrease, as shown in Figure 8b. Obviously, the turbulent kinetic energy of blown gas is negatively correlated with the NTA, which is consistent with the aforementioned analysis. A comparison of subfigures (a) and (d) in Figure 8 reveals great disparities in TKE caused by the NTA. Although the kinetic energy of blown gas is an important factor in oxygen top-blown converter steelmaking, the tangential velocity component of molten steel should also be taken into account. Therefore, we analyzed the flow field characteristics of the molten bath, as described in Section 3.2.

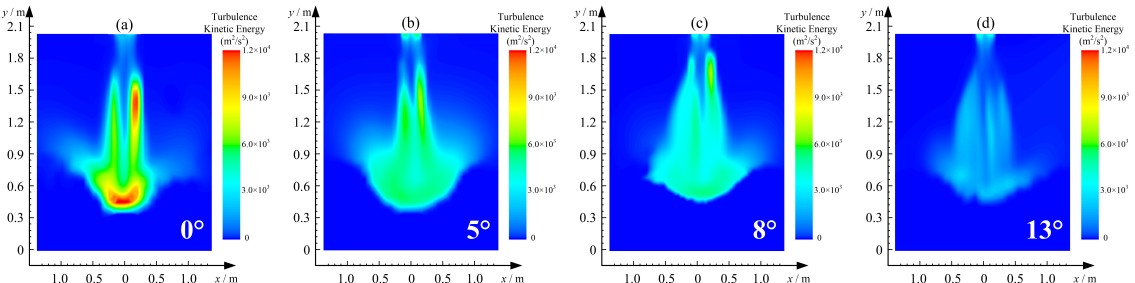

**Figure 8.** Turbulence kinetic energy profiles in top-blowing steelmaking with different multi-nozzle oxygen lances. (**a**) NTA = 0° (**b**) NTA = 5° (**c**) NTA = 8° (**d**) NTA = 13°.

### 3.2. Analysis of Stirring Effects and Impact Characteristics in Molten Bath Flow

The evolution of the gas–liquid phase in a steel converter is the main factor that leads to stirring and splashing of the molten bath. Most of the kinetic energy of the blown gas is lost at the gas–liquid interface, and the energy conversion of oxygen impacting the molten bath is ineffective. Accordingly, tracking the evolution of the two-phase interface is important for analyzing the flow field characteristics in a molten bath and characterizing the top-blowing state.

To explore the influence of the blown gas on the molten bath, we constructed a velocity vector diagram of the top-blowing with a conventional oxygen lance at the steel liquid level, as shown in Figure 9. In Figure 9a, the steel liquid near the wall had the maximum volume fraction and the center region had a minor volume fraction, which indicated that the shock effect range of the oxygen with high speed only focused on the center region of the molten steel. In addition, a great disparity is observed in the velocity of oxygen and molten steel. The gas flow rate is approximately 200–300 m/s, and the liquid steel flow rate is 0–3 m/s (Figure 9b). Therefore, the velocity vector diagram of molten steel is obtained by filtering the low velocity values (there may be some low-speed gas molecules) and filtering the high-speed molten steel, as shown in Figure 9c). To discuss the stirring effect in the molten bath, we need to analyze the velocity of various NTAs at different steel converter heights. Figure 10 shows the velocity vector (Y-plane projection) at the steel liquid level with different NTAs. In the surface layer, the velocity distribution and scale of 0°, 5°, and 8° oxygen lances are similar and all of them exhibit the phenomenon of high intermediate velocity and small surrounding velocity. However, the flow rate of the 13° oxygen lance is low, which is related to the large loss of kinetic energy in the blow process. In addition, the conventional and 5° nozzle form the vortex package in the area of the steel converter wall through several high-speed flow belts, resulting in molten flow at the surrounding area of the steel converter. The transition area of the 8° nozzle is annular, indicating that the stirring effects in the center and surrounding area are better; however, almost no flow occurs in the area near the wall. Two main reasons account for this phenomenon: (i) First, the tangential velocity component of the 8° nozzle is high, the molten steel exhibits stable swirling under the tangential force, and the peripheral velocity is relatively uniform because of the influence of the fluid viscous force. (ii) Second, the blow impingement of the conventional and 5° nozzles is intense and the local kinetic energy is high, which leads to a considerable impact depth. The splashing caused by impact increases the radial velocity of the molten steel, thus causing the aforementioned situation. To analyze the stirring effect of each NTA nozzle, we investigated the horizontal velocity at different molten steel depths, as shown in Figure 11. Although the conventional nozzle and 5° nozzle have advantages in some aspects, the 8°

oxygen lance has a good stirring effect in various layers of the molten steel. In addition, the 13° nozzle is less effective at swirling, consistent with the previous theoretical prediction.

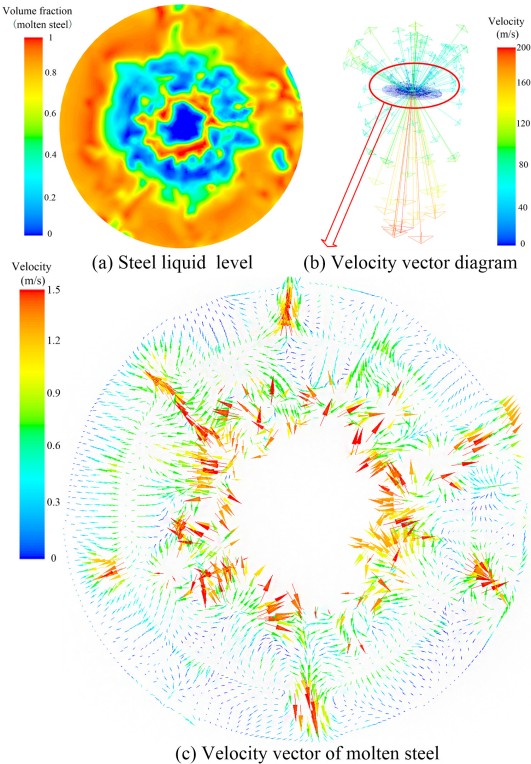

(a) Steel liquid level   (b) Velocity vector diagram

(c) Velocity vector of molten steel

**Figure 9.** Velocity vector diagrams of molten steel at the steel liquid level. (**a**) Volume fraction of molten steel at the steel liquid level. (**b**) Velocity vector diagram of gas–liquid interface. (**c**) Velocity vector diagram of molten steel.

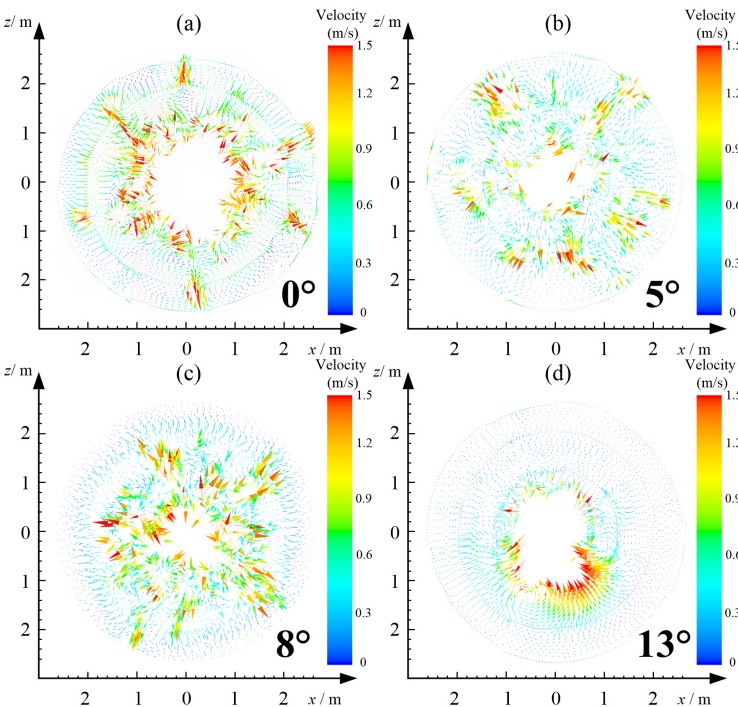

**Figure 10.** Velocity vector of molten steel at the liquid level with different multi-nozzle oxygen lances. (**a**) NTA = 0° (**b**) NTA = 5° (**c**) NTA = 8° (**d**) NTA = 13°.

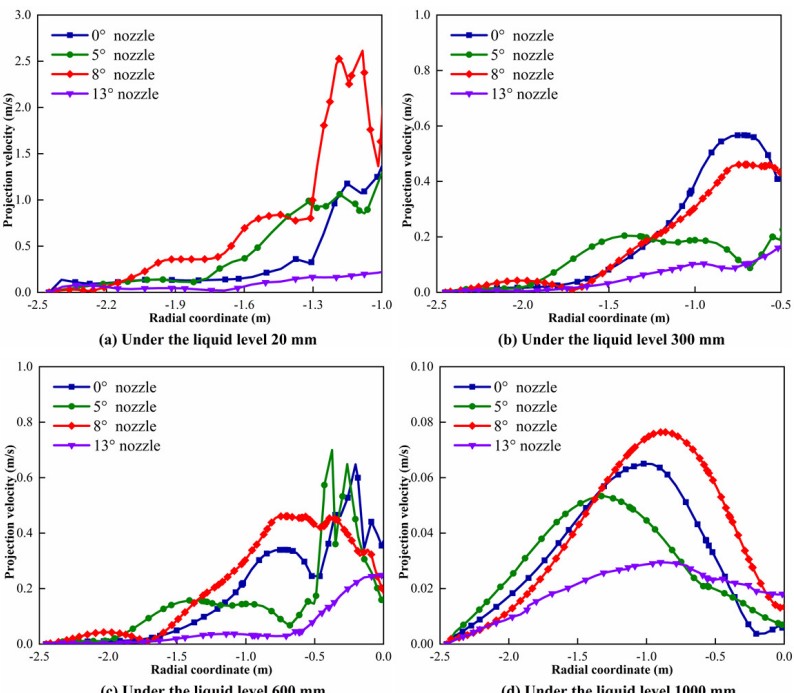

**Figure 11.** Molten steel movement in various heights with different multi-nozzle oxygen lances. (**a**) Under a liquid level of 20 mm. (**b**) Under a liquid level of 300 mm. (**c**) Under a liquid level of 600 mm. (**d**) Under a liquid level of 1000 mm.

Combined with the previous analysis, these results imply that the velocity of the molten bath is slow because of the premature dissipation of kinetic energy in the 13° oxygen lance, resulting in poor steelmaking efficiency. Therefore, in the design of a nozzle-twisted lance, the situations caused by excessive NTAs, such as jet interference, gas collision, and premature dissipation of kinetic energy, should be avoided. Compared with the conventional oxygen lance, the 5° nozzle has advantages in stirring effect at the liquid level. However, the swirl trend at the lower layer is not remarkable and the improvement is limited. The 8° oxygen lance induces a substantially improved stirring effect in the molten bath, but its effectiveness in the near-wall area is inferior to that of the conventional nozzle.

In the top-blow process, the supersonic gas impinges on the steel liquid level and a depression is formed in the molten steel by the huge kinetic energy generated. Meanwhile, the splash of molten steel in the converter poses a great threat to the nozzle. Therefore, reducing splash in the steel converter is beneficial to improving the service life of an oxygen lance. Figures 12 and 13 track the evolution mechanism of the two-phase interface between the conventional nozzle and the eight-degree nozzle in the preliminary stage of top blowing (the period when splash is most likely to occur). A comparison of these figures shows that the impact depth of the 8° oxygen lance is shallow and the splash is obviously weakened. This effect is mainly caused by the relatively low kinetic energy and the appropriate tangential velocity component. In addition, the simulation results are similar to the experiment of Higuchi et al. [10]. The results also suggest that the nozzle-twisted lance not only intensifies the local swirling in molten steel but also reduces the splashing of molten steel, which improves the efficiency of kinetic energy transfer and has substantial application prospects for steelmaking.

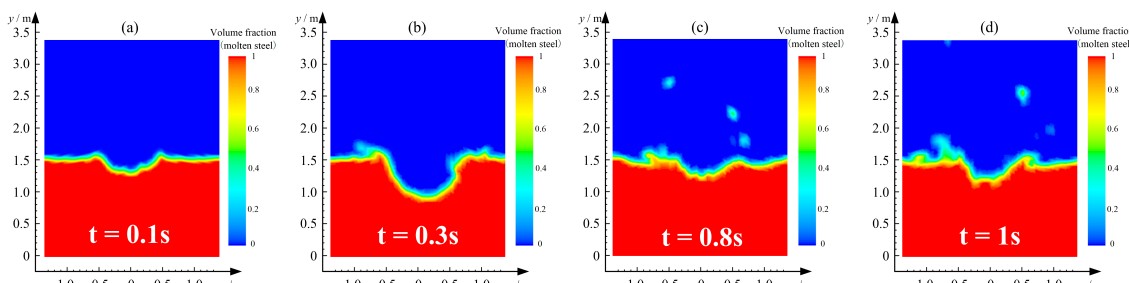

**Figure 12.** Spray of molten bath in the preliminary stage of top-blowing with a 0° nozzle twist angle (NTA) nozzle. (**a**) Flow time = 0.1 s, (**b**) flow time = 0.3 s, (**c**) flow time = 0.8 s, (**d**) flow time = 1.0 s.

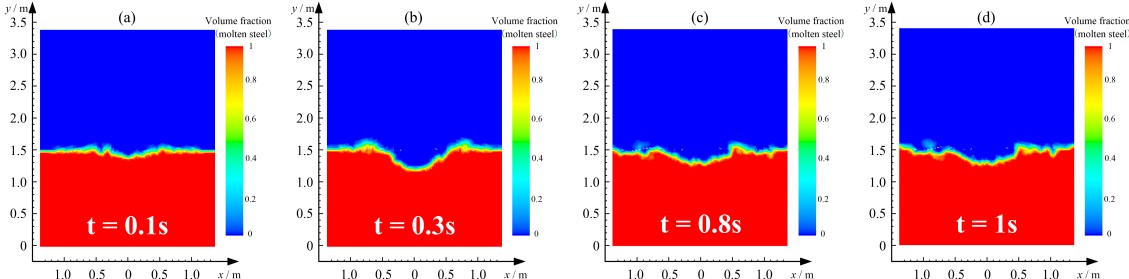

**Figure 13.** Spray of molten bath in the preliminary stage of top-blowing with an 8° NTA nozzle. (**a**) Flow time = 0.1 s, (**b**) flow time = 0.3 s, (**c**) flow time = 0.8 s, (**d**) flow time = 1.0 s.

## 4. Control Strategy for Multi-Nozzle Oxygen Lances

In the process of oxygen top-blown converter steelmaking, when the lance height is changed, the optimal NIA and NTA will also change. How to realize adaptive adjustment of the multi-nozzle oxygen lances' parameters depending on the different working conditions is unclear. Adaptive algorithms generally require training and adjustment based on a certain scale of data, which imposes higher requirements for production [31]. Numerical calculation can effectively fill this demand, and it can provide information about the flow field of the molten bath, such as the velocity and pressure, which cannot be determined experimentally [32]. Therefore, a coordinated numerical simulation and experiment is needed, where unavailable experiment data are provided by the numerical method to support training data for an adaptive control [33–35].

Based on the fuzzy control method, this paper attempts the optimal parameter adjustment control strategy between the lance height, nozzle inclination angle, and nozzle twist angle. The proposed control method firstly discretizes the three variables. Referred to actual working situation and above research, the lance height is discrete as 1200, 1600 and 2000 mm. In the present work, we propose an optimal parameter adjustment control strategy based on a fuzzy control method to control the lance height, NIA, and NTA. The proposed control method first discretizes the three variables. According to actual working situations and the aforementioned research, the lance height is discrete as 1200, 1600, and 2000 mm, the NTA is 0°, 5°, 8°, and 13°, and the NIA is 5.5°, 8.5°, and 13.5°. There are 36 different working conditions in total. Through the simulation of various working conditions, the advantages, and disadvantages of the stirring effect under each situation were obtained. The membership functions of the three variables and fuzzy rules were formulated; finally, the fuzzy control system for multi-nozzle oxygen lances was established.

Normalizing the vorticity of each height of molten steel can be regarded as the swirling performance under 36 working conditions. The domain of lance height is 1000–2200 mm and is divided into three fuzzy subsets: low height, middle height, and high height, respectively. The three subsets show a general bell-shaped distribution. The NTA and NIA domains are both 0°−15° and are divided into four and three fuzzy subsets, respectively. The three variable membership functions and Mamdani fuzzy system are shown in Figure 14a.

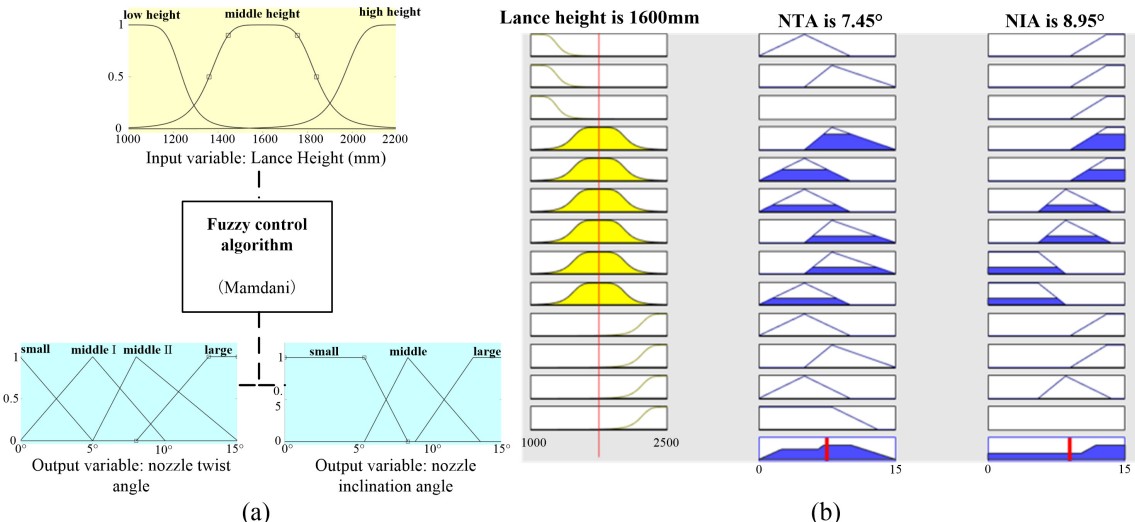

**Figure 14.** Fuzzy control system for multi-nozzle oxygen lances. (**a**) Three variable membership functions and Mamdani fuzzy system. (**b**) Fuzzy rule diagram of the control system.

Fuzzy rules are an important basis for fuzzy reasoning [36–38]. In this study, the fuzzy rules with different weights were formulated according to the optimal, sub-optimal, and the worst conditions under each lance height. Because of the small amount of data in the model, 13 fuzzy rules were established in this paper, as shown in Figure 14b. The fuzzy rules can be added, modified and deleted according to the actual situation in the industry field, and the inference curve based on the fuzzy control algorithm is shown in Figure 15. The parameter adaptive method based on the fuzzy control algorithm supplements the NTA and NIA well under different lance heights. The algorithm also indicates that the optimal NTA of the oxygen lance is within the range from 7.2° to 7.8°, which is similar to the results of Li et al. [6]. When the lance height is at 1000 –1400 mm, the optimal NIA is inversely proportional to the lance height. This is caused by the lance height, which has a significant influence on the impact kinetic energy, and the reduction of NIA is conducive to increasing the momentum blown into the molten pool. To maintain the jet angle and ensure the stirring effect, the optimal NTA will continue to rise in the process. When the lance height is within the range of 1400–1800 mm, the optimal NTA and NIA are relatively stable. It is caused by the lance height, which has little influence on the impact kinetic energy and the momentum in the process. With the continued increasing of lance height, the optimal NTA decreases, which is a consequence of the loss of kinetic energy when a large NTA nozzle blow is used and the remaining energy is insufficient to promote the molten bath [39,40]. This situation is consistent with the poor efficiency of the 13° oxygen lance in the previous analysis. Therefore, the control strategy based on the fuzzy control method has a good adaptability to optimize the parameters of multi-nozzle oxygen lances, and the results agree well with previous studies, theoretical analysis, and numerical simulations.

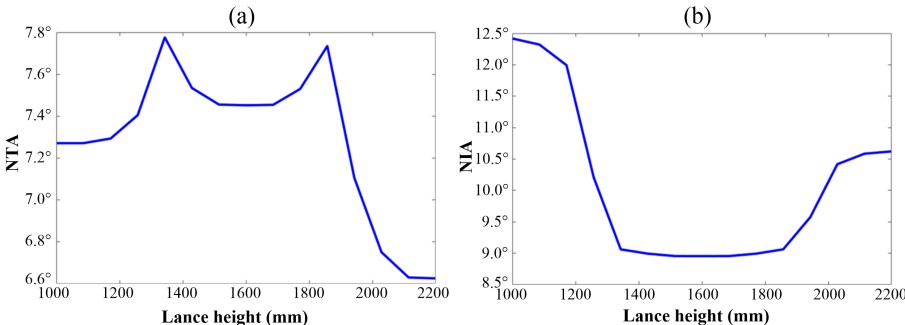

**Figure 15.** Inference curves based on the fuzzy control algorithm. (**a**) The optimal NTA variation with the lance height. (**b**) The optimal NIA variation with the lance height.

## 5. Conclusions

To address the low efficiency of the slagging reaction and energy transfer in the BOF process, we proposed an optimized scheme for the oxygen lance based on the CFD and fuzzy control method. The corresponding theoretical modeling and self-adjustment strategy were carried out to achieve the research goals, and the main conclusions are summarized as follows:

1. On the basis of the standard $k$-$\omega$ turbulence model and LSM, a fluid dynamics model for a BOF was set up and the four key factors (i.e., supersonic coherent jet behavior, stirring effect, energy transfer mechanism, slag–metal–gas emulsion) were obtained. The results show that the dynamic pressure and velocity of the conventional oxygen lance are the largest, and the impact depth is considerable. However, most of the kinetic energy is lost at the two-phase interface. In comparison, the coherent jet not only induces tangential velocity but also causes a decrease of the blown kinetic energy. Therefore, appropriate tangential velocity can improve the stirring effect and accelerate the slagging reaction. However, excessive cohesion will cause serious energy loss and cannot promote the slag–metal–gas emulsification.

2. To confirm the optimal design scheme for the slagging reaction, the stirring effect of different NTA oxygen lances was investigated. The comparison indicated that the 5° nozzle has apparent advantages over the conventional oxygen lance at the liquid level, but the improvement in the whole molten bath is not obvious. The 8° oxygen lance has a substantial stirring effect, but the near-wall velocity was weak. The results suggested that a uniform slagging reaction was achieved when the 8° oxygen lance was used. Otherwise, the optimal design scheme can improve the energy transfer efficiency and reduce the spray of the molten bath.

3. With respect to lance height, the self-adjustment control strategy for a multi-nozzle oxygen lance was adopted. The predictions indicated that the optimal NTA was within the range from 7.2° to 7.8°, and the superior NIA distribution from 9.0° to 12.5°. When the lance is at a low height, the optimal NIA is inversely proportional to the lance height. Within the lance height range of 1400–1800 mm, the optimal NTA and NIA are relatively stable. With the continued increasing of lance height, the optimal NTA decreases. It is in good agreement with previous studies, theoretical analyses, and simulations. The results presented here can provide support for saving and refining energy, improving top-blowing efficiency, and accelerating slagging reactions.

**Author Contributions:** D.T. and Z.Y. conceived and designed the research; Z.Y. and L.L. performed the simulations; Z.Y., T.W. and R.W. analyzed the data; J.L. and Y.H. contributed analysis tools and provided research platform; Z.Y., X.F., and H.L. wrote the paper. All authors have read and agreed to the published version of the manuscript.

**Funding:** This research was funded by the Natural Science Foundation of China (NSFC), grant number 51775501, and the Zhejiang Provincial Natural Science Foundation for Distinguished Young Scientists, grant number LR16E050001.

**Conflicts of Interest:** The authors declare no conflict of interest.

## Abbreviations

The following abbreviations are used in this manuscript:

| | |
|---|---|
| LSM | Level Set Method |
| CFD | Computational Fluid Dynamics |
| NTA | Nozzle Twist Angle |
| RANs | Reynolds Averaged Naiver–Stokes |
| FVM | Finite Volume Method |
| PRESTO | Pressure Staggering Option |
| CFL | Courant–Friedrichs–Lewy |
| NIA | Nozzle Inclination Angle |

| $d_t$ | Nozzle throat diameter |
|---|---|
| $d_e$ | Nozzle exit diameter |
| $L_n$ | Nozzle divergent length |
| $\theta$ | Nozzle inclination angle |
| $\alpha$ | Nozzle twist angle |
| $H_l$ | Oxygen lance height |
| $Ma$ | Design Mach number |
| $D$ | Steel converter diameter |
| $H$ | Steel converter height |
| $H_s$ | Steel liquid level |
| $\Phi$ | Generalized dependent variable |
| $S_\Phi$ | Generalized source term |
| $\boldsymbol{u}$ | Velocity vector |
| $\Gamma_\Phi$ | Generalized diffusion coefficient |
| $\mu_t$ | Eddy viscosity |
| $S_{ij}$ | Mean-velocity strain-rate tensor |
| $\delta_{ij}$ | Kronecker delta |
| $k$ | Turbulence kinetic energy |
| $\omega$ | Turbulence dissipation rate |
| $\rho$ | Fluid density |
| $l_1$ | The dispersed phase |
| $l_2$ | The continuous phase |
| $\Gamma$ | Boundary |
| $\phi$ | continuous dielectric surface |
| $\mu$ | Fluid density |
| $\tau$ | Virtual time |
| $d$ | Distance function |
| $\varepsilon$ | Modified value |

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
