# Peer review of "Optimized Scheme for Accelerating the Slagging Reaction and Slag–Metal–Gas Emulsification in a Basic Oxygen Furnace"

_applsci, doi:10.3390/app10155101_

Round 1
Reviewer 1 Report
Original research methods. Overall good work, and i would like to see how the practice verifies the results.
Author Response
Original research methods. Overall good work, and I would like to see how the practice verifies the results.
Response: Thanks for your careful reading and beneficial advice. We set up a fluid dynamics model for jet dynamics behaviour to discuss the stirring effects of top-blown. Based on the simulations in various working conditions, we predict the optimal NIA and NTA at different lance height by fuzzy algorithm. We hope our research works not only can offer theoretical references to the research works of BOF steelmaking, but also can provide technical support and direct guidance for the design work of multi-nozzle oxygen lance. For the awkward problem of English language appearing in the manuscript, we have asked an English language editor to help us, and made some modifications to the paper.
Aiming at practical problems, we refer the research of Sun et al in detail. They used a modified lance in PANSTEEL and achieve significant improvements, in which the average time of early slag formed was decreased from 5.25 to 4.1 min, the rate of dephosphorisation was increased from 80.02% to 82.49%, and the life of oxygen lance is raised from 428 heats to 800 heats (highlighted in line 52). In this research, the fuzzy prediction results reflect the simulation conclusions and are consistent with the conclusion of other scholars, which also reflects the effectiveness of the method. To provide more support for the fuzzy method, our current work is to establish the water model experiment platform and simplify the simulation process. The latter paper will focus on the practicability of fuzzy control strategy to predict the optimal parameters of oxygen lance.
We would improve the manuscript continuously, and wish it could match the publication requirements of Applied Sciences.
Reviewer 2 Report
The article discusses a very important issue in the steel making. Improving the efficiency of BOF is an important factor as improving the whole steel making production chain. The proposed methods seem to be sound and they have a good potential to have significance to improve the BOF-process.
The methods are well explained in the article and used CFD-method and its results are analyzed. The fuzzy control strategy is introduced and developed, however there is no results of using it. Possibly it will be introduced in a latter paper?
The conclusions are based on the theoretical analyses and results from CFD-model and they indicate that improvement on the BOF-process would be possible to achieve. It would be very interesting to see the results achieved from the actual plant or at least from water-model. The improvements achieved from the actual steel plant can be tricky to address as the variation in the production is large.
The paper with the improvements can be a good asset into the scientific literature.
Couple corrections as an example:
Consider changing a verb ”jet” with a verb ”blow”.
Line 38: most losses at the slag-metal-gas emulsions à rephrase to be easier to understand
If you use word “converter” instead of BOF, use “steel converter” as word converter refers primarily to electrical engineering.
Change all “et al.” to “et al.”(in Italic form)
Rephrase two sentences from line 49 to 52 into an easier form to understand. Starting “Sun et al.”
Line 52: jet process is an important dynamics behavior to trigger --> jet process’s dynamics has an important role to trigger
Line 99: multi-nozzle oxygen lances was studied which designed in MASTEEL --> multi-nozzle oxygen lances designed in MASTEEL were studied
Line 103: For improve --> To improve
Line 245: The fluid sloshing --> maybe more common terms: splashing, slopping (mostly slag) or spitting (mostly iron). Even though this is CFD-calculation, maybe it is good to use terms used in the steel making.
The grammar of the paper needs to be corrected by a competent person.
Author Response
Dear Reviewer,
Thanks for your careful reading and beneficial advice. Ours response has been uploaded in the attachment.
Zichao Yin, Jianfei Lu*, Lin Li, Tong Wang, Ronghui Wang, Xinhua Fan, Houkai Lin, Yuanshun Huang, Dapeng Tan*.
15, July, 2020.

Reviewer 3 Report
The authors deal with simulations, optimization, and control for the multi-nozzle oxygen lance; they showed optimal NTA angle ranges and NIA distributions. The results could be supportive for the further development of multi-nozzle oxygen lance.
Author Response
The authors deal with simulations, optimization, and control for the multi-nozzle oxygen lance; they showed optimal NTA angle ranges and NIA distributions. The results could be supportive for the further development of multi-nozzle oxygen lance.
Response: Thanks for your positive comments. We set up a fluid dynamics model for jet dynamics behaviour to discuss the stirring effects of top-blown. Based on the simulations in various working conditions, we predict the optimal NIA and NTA at different lance height by fuzzy algorithm. We hope our research works not only can offer theoretical references to the research works of BOF steelmaking, but also can provide technical support and direct guidance for the design work of multi-nozzle oxygen lance. For the awkward problem of English language appearing in the manuscript, we have asked an English language editor to help us, and made some modifications to the paper.
We would improve the manuscript continuously, and wish it could match the publication requirements of Applied Sciences.
Reviewer 4 Report
This was an interesting paper that was well documented. There are a few areas that require improvement.
--English grammar and paper structure throughout the paper needs to be improved (i.e. structure of sentences, plurals, spelling, consecutive sentences starting with the same word, spacing, capitalization, use of "we", do not start a sentence with AND, etc) .
--Proper reference to PANSTEEL
--Discuss individual references, please do not lump references.
--Are all the variables properly defined? Maybe a nomenclature is better to include
--How do you arrive at the constants in lines 123-124?
--Be careful and check all your equations. It is confusing I think eq 11 and 12 are the same...why?
-- is the proper function SGN not sign for sign? eq 16 and line 160
--please discuss accuracy and error
--Many of the figures are to small to be meaningful (i.e. fig 9, 10, etc)
--line 241-244 (I) should be (i), (ii), etc
--more detail and explain fig 15..why in (a) does it drop off and then rise?
--line 309 spelling error
---- line 318 spacing
--conclusions and results should be more detailed
--please update ref 11 and 12 2019 in press. They should be accepted or rejected by now.
Author Response

(The authors gave the same response as above.)

Round 2
Reviewer 4 Report
Where are the various Schemes defined?
More descriptive figure titles are required (i.e. Figure 4, 12,13, etd has several parts that require details in the title, etc)
Each figure and part of figure needs to be discussed in the text. This needs to be done throughout
There are still lumped references that can be discussed individually (i.e. [1-3], [9,10],etc) please revise throughout
line 64 reference for Li not provided...Others throughout as well
Reference to PANSTEEL, MSSTEEL still not provided throughout
English grammar still needs review throughout (i.e. Consecutive sentences starting with the same word "The" Lines 106-114, Etc)
Spacing between units and number, after a . Needs to be chacked and corrected throughout
Please provide a more thorough discussion about accuracy
Some figures are still small and difficult to obtain information (i.e fig 9, etc)
Author Response
Dear Reviewer, Thanks for your careful reading and beneficial advice. Ours response has been uploaded in the attachment.We would improve the manuscript continuously, and wish it could match the publication requirements of Applied Sciences. Zichao Yin, Jianfei Lu*, Lin Li, Tong Wang, Ronghui Wang, Xinhua Fan, Houkai Lin, Yuanshun Huang, Dapeng Tan*. 18, July, 2020.
